# Structural and Comparative Analyses of Insects Suggest the Presence of an Ultra-Conserved Regulatory Element of the Genes Encoding Vacuolar-Type ATPase Subunits and Assembly Factors

**DOI:** 10.3390/biology12081127

**Published:** 2023-08-13

**Authors:** Domenica Lovero, Damiano Porcelli, Luca Giordano, Claudio Lo Giudice, Ernesto Picardi, Graziano Pesole, Eugenia Pignataro, Antonio Palazzo, René Massimiliano Marsano

**Affiliations:** 1Dipartimento di Bioscienze Biotecnologie e Ambiente, Università Degli Studi di Bari Aldo Moro, Via Orabona 4, 70125 Bari, Italy; domenica.lovero@masmecbiomed.com (D.L.); damiano.porcelli@gmail.com (D.P.); ernesto.picardi@uniba.it (E.P.); graziano.pesole@uniba.it (G.P.); eugenia.pignataro@uniba.it (E.P.); antonio.palazzo@uniba.it (A.P.); 2MASMEC Biomed S.p.A., Via Delle Violette 14, 70026 Modugno, Italy; 3METALABS S.R.L., Corso A. De Gasperi 381/1, 70125 Bari, Italy; 4Cardio-Pulmonary Institute (CPI), Universities of Giessen and Marburg Lung Center (UGMLC), Member of the German Center for Lung Research (DZL), Justus-Liebig-University, Aulweg 130, 35392 Giessen, Germany; luca.giordano@innere.med.uni-giessen.de; 5Istituto di Tecnologie Biomediche (ITB), Consiglio Nazionale Delle Ricerche, Via Giovanni Amendola, 122, 70126 Bari, Italy; claudio.logiudice@ba.itb.cnr.it

**Keywords:** V-ATPase, lysosomal biogenesis, gene regulation, dCLEAR, evolution, *cis*-regulatory sequences, Drosophila, insect, gene duplication

## Abstract

**Simple Summary:**

The study of whole-genome sequences, combined with advanced bioinformatic tools, has revolutionized our understanding of the evolutionary history of genes and genomes. Here, we have highlighted the conservation of a putative regulatory element in insect genes encoding for an important enzymatic complex of the vacuolar and lysosomal compartments of the eukaryotic cell with storage and recycling functions, respectively. This study will enable future investigations aimed at understanding regulative circuits that generate cellular complexity.

**Abstract:**

Gene and genome comparison represent an invaluable tool to identify evolutionarily conserved sequences with possible functional significance. In this work, we have analyzed orthologous genes encoding subunits and assembly factors of the V-ATPase complex, an important enzymatic complex of the vacuolar and lysosomal compartments of the eukaryotic cell with storage and recycling functions, respectively, as well as the main pump in the plasma membrane that energizes the epithelial transport in insects. This study involves 70 insect species belonging to eight insect orders. We highlighted the conservation of a short sequence in the genes encoding subunits of the V-ATPase complex and their assembly factors analyzed with respect to their exon-intron organization of those genes. This study offers the possibility to study ultra-conserved regulatory elements under an evolutionary perspective, with the aim of expanding our knowledge on the regulation of complex gene networks at the basis of organellar biogenesis and cellular organization.

## 1. Introduction

The increased availability of whole-genome sequences, coupled with the development of new analytical tools, has significantly enhanced our understanding of the evolutionary history of genes and genomes. This advancement provides valuable insights into the structural and functional features of extant genomes [1,2,3,4,5]. By combining experimental data with computer-aided genome comparisons across evolutionarily distant species, researchers have successfully identified the structure of protein-coding genes [6], identified nonfunctional genes [7], and discovered conserved non-coding regions that likely have functional significance. These regions may contain motifs involved in the coordinated expression of a set of genes encoding subunits of enzymatic complexes or genes correlated by function [8].

Co-regulated gene networks can be effectively studied by examining sets of genes encoding core proteins of essential complexes in specific subcellular compartments. The mitochondrial and chloroplast compartments serve as excellent models for studying co-regulated genes [9].

An alternative and complementary approach to uncovering gene regulatory networks involves the identification of over-represented conserved DNA motifs in specific gene regions. Similar analyses unveiled the possible co-regulated expression of chloroplast and nuclear genes encoding chloroplast proteins [9] and the ribosomal proteins coding genes [10].

Of particular interest are the vacuolar and lysosomal compartments, which harbor essential and evolutionarily conserved multi-subunit complexes. Genes encoding the subunits of the vacuolar (V-type) ATPase complex present an excellent candidate model for studying co-regulation in gene networks.

V-ATPases are ATP-dependent proton pumps involved in the acidification of eukaryotic intracellular compartments. Their functions extend beyond cellular pH regulation, as they play vital roles in various cellular processes that depend on cell type and intracellular localization [11,12]. Through lysosomal acidification, the V-ATPase participates in protein degradation and in the control of endosomal trafficking and sorting [13,14]. Given their critical role in cellular function, V-ATPases impact many physiological processes, including male fertility [15,16], bone homeostasis [17], and urine formation [18], and are implicated in the pathophysiology of various human diseases [19]. They are implicated in pathogen-borne diseases like influenza [20] and SARS-CoV-2 infections [21], as well as genetic diseases such as deafness [22] and neurodegenerative diseases [23]. The V-ATPase is present in virtually all epithelial cells of insects, and it accomplishes various physiological functions such as ion transport and balance, nutrient absorption, and waste excretion [24,25,26]. These processes require the active transport of ions across the epithelial membrane, and V-ATPase serves as a key player in this energization process [27]. Regulation of the V-ATPase by V0-V1 dissociation has been examined most extensively in insect non-model systems, such as in *Manduca sexta* midgut during molting [28] and in the blowfly salivary glands [29]. Given its evolutionary conservation, V-ATPase has been identified as an attractive target for the development of pest management. A wide range of severe phenotypes, ranging from developmental abnormalities [30,31] and developmental stop [32,33] to the drop in the fertility rate [34] to lethality [35,36], have been observed in a broad range of insects where the V-ATPase-encoding genes were knocked down. These observations also highlight the vital role of the V-ATPase complex.

The basic structure of V-ATPases consists of two functional domains: V_1_ and V_0._ The V_1_ domain, composed of eight different subunits (A_3_, B_3_, C_1_, D_1_, E_1_, F_1_, G_2_, H_1_) [37], is located on the cytoplasmic side of the membrane and binds and hydrolyzes ATP, providing the energy for proton translocation across the membrane-embedded V_0_ domain.

The V_0_ domain contains five different subunits in a possible stoichiometry of a_1_ b_1_ c_6_ d_1_ e_1_ [20]. Two accessory proteins (AC45 and M8.9) are also associated with the V-ATPase complex. Whereas most of the 15 subunits have been shown to be essential structural and functional components of the complex, the precise function(s) of many of them remain undetermined. Moreover, in higher eukaryotes, several V-ATPase subunits are known to have multiple isoforms that can be encoded by different genes or expressed by alternative splicing contributing to generating complex expression patterns in different tissues [38,39,40,41,42,43]. In the genome of *Drosophila melanogaster*, 33 genes encode the 15 subunits of the V-ATPase [44]. Only five subunits of the V_1_ domain (B, C, E, G, and H) and the two accessory subunits (AC45 and M8.9) are encoded by single genes, whereas the other subunits are multi-isoform and are expressed by two or more genes. However, the subunit composition of the holoenzyme responsible for the basic V-ATPase function in the major transporting epithelia of the fruit fly has been exactly defined by ESTs analysis and by microarray and in situ data [44].

While the functional significance of the V-ATPases isoform complexity is at present not fully understood, it is likely that different enzyme isoforms are required for tissue-specific control of V-ATPase expression [45].

It has been previously observed that coordinated regulation of the gene networks is required for lysosomal biogenesis in human cells [46]. Many lysosomal genes are regulated by the transcription factor TFEB [47] that recognizes and binds to a cis-regulatory element called CLEAR (Coordinated Lysosome Expression and Regulation), thus promoting the transcription of target genes. This finding might have important consequences in the development of alternative therapeutic strategies, especially in neurological diseases [48].

A similar regulation has been observed in Drosophila melanogaster, where a homologous *cis*-regulatory element has been identified in association with genes encoding V-ATPase subunits and regulatory factors [8,49]. This motif is recognized by MITF, an evolutionarily conserved transcription factor homologous to TFEB [50], which can bind to the dCLEAR motif of the V-ATPase coding genes.

In this work, we have analyzed a set of orthologous genes encoding subunits of the V-ATPase complex and their assembly factors identified in the genomes of 70 species representative of 8 Insect Orders. The comparative analysis of these genes uncovered high conservation of the dCLEAR element position with respect to the V-ATPase gene structure in insects, suggesting the existence of an ultra-conserved gene regulation system in insects with a striking similarity to the CLEAR/TFEB-based regulation system observed in vertebrates.

## 2. Materials and Methods

The genomic sequences analyzed in this work were retrieved from the Flybase [51] (http://flybase.org; accessed on 1 April 2023) VectorBase [52] (https://www.vectorbase.org/, accessed on 1 June 2020), and NCBI databases [53] (https://www.ncbi.nlm.nih.gov/, accessed on 1 May 2023).

Using the initial set of query sequences (Appendix A), we used BlastN and TBlastN [54] to search orthologous genes in the species analyzed in this work. The top-scoring hits were selected and further analyzed. The complete list of species and the accessions in which the V-ATPase genes analyzed in this work were found is reported in Appendix A.

A BLAST search was also used to infer exon-intron structure if mRNA sequences were available. Otherwise, the exon-intron structure was inferred using the GeneWise tool [55] https://www.ebi.ac.uk/Tools/psa/genewise/ (accessed on 17 July 2023).

Multiple alignments were obtained using the MultAlin 5.4.1 software [56] at the MultAlin server (http://prodes.toulouse.inra.fr/multalin/multalin.html, accessed on 17 July 2023), using the DNA 5-0 scoring matrix.

To identify dCLEAR motifs, we used the Regulatory Sequence Analysis Tools from the RSAT server (http://rsat.sb-roscoff.fr/, accessed on 17 July 2023) [57]. The graphical representations of dCLEAR motifs as sequence logos [58] were generated using WebLogo (https://weblogo.berkeley.edu/logo.cgi, accessed on 17 July 2023) [59].

Gene expression data were obtained from FlyAtlas (accessed on 1 May 2023) [60] and from BeetleAtlas (accessed on 17 July 2023) [61]. Heatmaps have been constructed using HeatMapper (http://www.heatmapper.ca/, accessed on 17 July 2023) [62].

Phylogenetic relationships between the species studied in this manuscript were obtained from TimeTree (http://www.timetree.org/, accessed on 17 July 2023) [63] and edited with FigTree v1.4.4 (http://tree.bio.ed.ac.uk/software/figtree/, accessed on 17 July 2023).

## 3. Results

We investigated a set of V-ATPase subunits encoding genes of *D. melanogaster* (Appendix A), including the fifteen V-ATPase subunits encoding genes that are responsive to *Mitf1* previously identified [8]. As *CG5969* is upregulated upon MITF overexpression [8], we hypothesized that V-ATPase assembly factors might be regulated by the same transcription circuit. To explore this possibility, we conducted a matrix scan analysis revealing that both *CG5969* and *CG7071* contain a dCLEAR element within or overlapping the 5′UTR region. Specifically, the dCLEAR element is located exactly 5 bp upstream of the TSS of *CG5969* and 275 bp downstream of the TSS of *CG7071*. Consequently, we included in our analyses two additional genes (*CG5969* and *CG7071*) encoding V-ATPase assembly factors homologous to the human *Vma21* and *Tmem199* assembly factors, respectively. Using this extended dataset, we searched for orthologous genes encoding V-ATPase subunits in 23 Drosophilidae species, and subsequently, we expanded the dataset to 72 Artrhopoda species (Appendix A). This expanded dataset encompassed forty-three species of Diptera (twenty-three species of Drosophilidae, fourteen species of Culicidae, two species of Psychodidae, one Glossinidae, one Tephiridae, one Muscidae, one Cecidomyiidae), five species of Lepidoptera, one Strepsiptera, two Coleoptera, fifteen Hymenoptera, one Odonata, two Hemiptera, one Phtiraptera, one Crustacea, and one Ixodida. Additionally, *Homo sapiens*, *Nematostella vectensis,* and *Trichoplax adhaerens* were included in the study as outgroups.

A simplified evolutionary tree including all species studied in this work is shown in Figure 1, in which it should be highlighted that the species examined in this study cover a very large evolutionary time (roughly 550MY divergence Ixodes vs. Drosophila).

We annotated a set of 1750 genes encoding V-ATPase subunits and assembly factors, enabling us to investigate the evolution of gene structure, and the presence and conservation of the cis-acting motif that is known to play a crucial role in the regulation of this group of coordinately expressed genes.

### 3.1. The dCLEAR Element Is Found in a Subset of the Genes Encoding V-ATPase and Their Assembly Proteins

We employed a DNA pattern-discovery approach to identify over-represented DNA motifs in the initial set of genes encoding the 17 proteins involved in the V-ATPase biogenesis of *D. melanogaster*. The top-scoring matrix (rank1 in the RSAT output) highlighted an 8 bp motif TCACATGA/TCATGTGA, which is identical to the core binding site described in Zhang et al. [8] (Figure 2A). We will refer to this motif as dCLEAR (for Drosophila CLEAR [49]) throughout this manuscript.

A pattern matching analysis using the matrix that describes the dCLEAR against all the 17 sequences set suggests that it is enriched in the 5′ regions of all these genes, in a sequence range covering 200 bp around the TSS (−50 to +150) and more distantly in a range between 300 and 1200 bp downstream the TSS (Figure 2B).

As additional proof of the enrichment of the dCLEAR motif in the V-ATPase genes, we have detected only 161 occurrences of the dCLEAR motif in 1000 random sequences retrieved from *D. melanogaster*.

Assuming a random distribution of the dCLEAR motif within the transcriptional unit, its occurrence is expected at the same frequency in UTRs, exons, and introns. However, the dCLEAR element is significantly enriched in the first intron of the analyzed genes (χ^2^ = 15.25, *p* = 0.00420982, df = 4). A detailed representation of the dCLEAR position within each V-ATPase-related gene analyzed in this work is shown in Figure 2C.

### 3.2. The dCLEAR Motif Is Strongly Conserved in Drosophilidae

The matrix describing the dCLEAR motif in *D. melanogaster* was also used to identify matches in the V-ATPase coding genes of other Drosophila species. Out of 779 V-ATPase coding genes tested in Drosophilidae, 385 genes contain the dCLEAR motif in a homologous region compared to *D. melanogaster*. Figure 3A provides an example of sequence and positional conservation of the dCLEAR motif in 23 Drosophila species. It demonstrates the presence of the dCLEAR elements in the first intron (two dCLEARs) and in the second intron (six dCLEAR) of the *Vha16-1* gene. In all the species of Drosophilidae compared in Figure 3A, it can be observed a high degree of sequence and positional conservation of their respective dCLEARs. In a few species, some of the dCLEAR has not been identified in the *Vha16-1* gene (i.e., *D. suzukii*, *D. persimilis*, *D. willistoni*, and *D. albomicans*). A careful inspection of the dCLEAR in the V-ATPase genes of Drosophilidae suggests that the most distal dCLEAR elements are more prone to be lost, while the most proximal is probably most constrained (see Appendix A). This strong conservation pattern has been observed in all the genes listed in Appendix A, suggesting that dCLEAR has acquired a functional role early in the evolution of Drosophilidae.

In order to obtain a descriptor of the dCLEAR motif that could be suitable for the Drosophilidae family, 832 homologous dCLEARs in 385 genes were aligned and used to build a new comprehensive matrix and logo (Appendix A), which is consistent with that described in previous works [8,49]. This finding further supports the evolutionary conservation of the dCLEAR motif.

It is worth noting that the dCLEAR motif is exclusively found in a subset of V-ATPase coding genes that are putatively orthologous to the Mitf-regulated genes in *D. melanogaster* [8] (Appendix A). More intriguingly, considering the evolutionary events that have generated paralogous genes in many species (better described later in the text), only those genes having a tissue-specific expression pattern retain the dCLEAR elements (see Section 3.4). This observation suggests that the Mitf:dCLEAR interaction has been preserved during the evolution of the Drosophila species analyzed, and it is connected to the generation of complex gene expression patterns.

The positional analysis of the dCLEAR element in the V-ATPase-related genes in the genome of Drosophilidae species suggests that it is preferentially located at the 5′ end of the analyzed genes. Out of 720 elements, 318 can be unambiguously mapped at the 5′ of the genes analyzed, other nearby upstream or downstream the TSS, whereas additional 178 dCLEAR elements can be considered either at the 5′ of alternative transcriptional isoforms or in the intron of the uppermost transcribed isoforms.

In conclusion, as shown in Appendix A, the dCLEAR position is strongly conserved in orthologous V-ATPase genes in Drosophilidae.

### 3.3. The dCLEAR Motif Is Conserved in Insects

The dCLEAR motif search in orthologous genes identified in the non-Drosophilidae Diptera (Lepidoptera, Hymenoptera, Coleoptera, Strepsiptera, and Odonata) has been performed using either the Drosophilidae-specific dCLEAR matrix, or Order-specific matrices (Appendix A). It is evident that in most cases, the position of the dCLEAR motif is maintained within the same region (5′/5′UTR or intron), although additional occurrences can sometimes be found in downstream introns. As an example, the comparison of the dCLEAR motifs identified in the set of orthologous genes encoding the Atpv0c subunit in insects is shown in Figure 3B. A striking conservation of the dCLEAR motif in species of the same family (e.g., Culicidae) or at the Suborder level (Lepidoptera, Glossata) was observed.

We also observed that the two central nucleotides of the motif can slightly vary between the different Orders (Appendix A).

*M. moldrzyki* (Strepsiptera) is the only insect species in which the dCLEAR motif was not found in any of the V-ATPase genes analyzed in this study.

### 3.4. V-ATPase Gene Duplications in Drosophilidae

To gain insights into their evolutionary origin, we analyzed duplicated genes in detail. The genomic organization of many V-ATPase duplicates suggests that they likely originated from retroposition events, as indicated by their intron-less structure, or the presence of fewer introns compared to their paralogs. In other cases, duplicate genes maintain an identical or very similar exon/intron structural organization suggesting that they are the result of either duplication, transposition, or recombination events occurring at the genomic region spanning the parental gene locus. Gene duplication, regardless of the underlying mechanism, has occurred frequently in the evolutionary history of the V-ATPase genes in Arthropoda.

A careful analysis of the V-ATPase duplicates in Drosophilidae, along with the positional analysis of the associated dCLEAR elements (if present), allow us to depict three possible scenarios to describe the origin of duplicate genes based on the available data (Figure 4).

In the first scenario (Figure 4A), the ancestral gene underwent either duplication or retroposition. The parental copy retained the dCLEAR element, and possibly maintained the original function (*Vha68-2*, *Vha14-1*, *Vha100-2*, *Vha16-1*, *Vha39-1*, *VhaM9.7-b*), while the duplicated gene may have lost the dCLEAR element resulting in a sub-functionalized gene with a testis-specific expression pattern (*Vha14-2*; Figure 5A) or it may have acquired a tissue-specific expression pattern (*Vha-68-1* and *Vha-100-1*). Similarly, the retroposed copy may have acquired either a testis-specific expression pattern (*Vha68-3*, *Vha100-3*, *Vha16-2*, *Vha16-3*, *Vha16-4*, *Vha16-5*, *Vha39-2*, *VhaM9.7-d*) or it may have evolved into a tissue-specific paralog (*VhaM9.7-a*, *VhaM9.7-c*, *Vha100-4*, *Vha100-5*). It is important to note that partial reverse transcription, which can lead to 5′-truncated cDNAs, can explain the loss of the dCLEAR element when it is located in the 5′UTR of the original gene copy. Indeed, this is the case for *Vha39-2* and *VhaM9.7-d* as they have a shorter 5′UTR compared to the original copy (*Vha39-1, VhaM9.7b*, respectively). In other cases, loss of the dCLEAR motif upon retrotrascription can be due to its presence in introns.

In an alternative scenario, only retroposition accounts for the generation of paralogs (Figure 4B). As observed for the *Vha36-3* gene, the ancestral gene likely lost its dCLEAR element(s), while the newly generated copy gained one or more dCLEAR elements to become a functional gene.

On the other hand, the *VhaPPA1* ancestral gene was probably lost early in the evolution, before the diversification of Diptera and Lepidoptera (see Appendix A); the retroposed paralog has either acquired a new dCLEAR, hence becoming the functional ortholog (*VhaPPA1-1*), or sub-functionalized in testes (*VhaPPA1-2*; Figure 5A,B).

The third scenario (Figure 4C) depicts the evolutionary history of the *Vha14* gene (Atp6v1F), which has undergone species-specific changes.

The comparison of the genes encoding the Atp6v1F subunit in non-drosophilid species suggests that the ancestral structural ortholog had four exons, with the dCLEAR element located at the 5′ of the transcriptional unit (Appendix A). We hypothesize that a duplication event that pre-dates the split of all the Drosophila species generated a sub-functionalized copy featured by the loss of the dCLEAR element. Subsequently, a retroposition event that precedes the split of the obscura and melanogaster groups led to the replacement (possibly by homologous recombination) of the parental gene, while it is still detectable in the *D. willistoni*, *D. albomicans*, *D. virilis*, *D. grimshawi*, and *D. mojavensis* species. The retroposed-recombined copy retained the original dCLEAR element at the 5′UTR of the gene, while in the original copy, the dCLEAR motif degenerated over time.

The presence of three copies of the ATP1F gene in *D. virilis* suggests a more entangled evolutionary history (Figure 6). Part of the inferred evolutionary history in *D. virilis* recalls the scenario described above. Indeed, the first duplication event gave rise to two copies (Figure 6, A and B copies), one of which (*Dvir\GJ14239*) lost the dCLEAR element (Figure 6, B’ in). The other copy underwent additional duplication (A and C copies, Figure 6). One of these copies (*Dvir\GJ10919*) was replaced by a retroposed copy (depicted as C’ in Figure 6), whereas the other one (*Dvir\GJ22513*) retained the structure and possibly the function of the ancestral gene (shown as the structural and functional ortholog in Figure 6).

Formally, the gene replacement event described above represents a case of intron exclusion through cDNA-directed homologous recombination, which has already been described in *D. melanogaster* [64,65]. Intron exclusion is an important evolutionary process in which homologous recombination events occur between the cDNA generated via retroposition and its genomic, intron-containing locus, resulting in the generation of an intron-free gene. These recombination events are likely triggered by DNA double-strand breaks and can lead to the loss of some or all of the introns contained in the parental gene.

Other possible examples of intron exclusion detected this study are represented by the *Dkik\Vha68-2*, *Dkik\Vha55*, *Dwil\Vha44* (*GK21692*), *Dwil\Vha36-2* (*GK22511*), *Deug\Vha100-2*, *Dana\GF18112*, *Dbip\VhaM9.7-b2* (Appendix A, respectively).

### 3.5. Evolution of the Exon-Intron Structure of the V-ATPase Coding Genes in Insects

Selection operates on intron sequences because they may either contain ORFs or may form part of the mature mRNA due to alternative splicing. Introns also play a regulatory role in gene expression or are involved in maintaining pre-mRNA secondary structure [66]. However, the evolutionary mechanisms and dynamics of intron gain and loss are only partially understood. Based on the assumption that comparative analyses of the structural organization of V-ATPase encoding genes in an informative range of species would prove useful to study the mechanisms that shape gene structure during evolution, we first compared the exon-intron structure of V-ATPase orthologous genes in 23 Drosophilidae species; then we assessed intron gain/loss in the orthologs of these genes in non-Drosophilidae insects.

The comparative analysis of the gene structure in each insect’s gene cluster is reported in Appendix A). Looking only at the number of introns that interrupt the coding exons, there is a pronounced progressive loss of introns across arthropods, with an intron-rich gene structure in the basal Classes (Arachnida and Brachiopoda) and a dramatic reduction in intron numbers in Diptera.

As an illustrative example, the comparison of genes encoding the ATP6v1H subunit is shown in Figure 7. Assuming that the basal metazoan *N. vectensis* and *T. adhaerens* have retained the ancestral gene structure, the evolutionary history of the ATP6v1H-encoding genes structure can be traced back, considering that an intron loss event results in the fusion of the two adjacent exons, and that an intron gain event results in exon fission. It is important to note that these events are not expected to significantly alter the overall length of the coding sequence.

Intron loss has occurred repeatedly in the analyzed lineages (Figure 7). Yet, in *H. sapiens*, it is evident that five introns (namely introns 4, 8, 10, 11, 13) have been lost compared to the two basal metazoans. An intron gain event occurred early in the evolution of Arthropoda (see *I. scapularis*). This newly gained intron, located at the 5′ of the coding region, has been maintained in all the analyzed Arthropoda species, and only in some Dipteran species has it been lost. Interestingly, Culicidae and Coleoptera have evolved a dCLEAR sequence in this new intron. In Hymenopteran, the ATP6v1H-encoding genes have lost an additional intron (intron 12). However, the ancestral line pre-dating the Lepidoptera/Diptera/Strepsiptera/Coleoptera split gained three new introns in this gene region. These introns have been completely (see *L. longipalpis* and *P. papatasi*, Family Psychodidae) or partially (Family Muscidae, Culicidae, Glossinidae, Tephritidae, and Drosophilidae) lost in Diptera. It is worth noting that each intron loss is followed by the fusion of adjacent exons resulting in a new exon whose length is, in most cases, the exact sum of the ancestral exons.

In Drosophilidae, the exon-intron structure of the V-ATPase genes is nearly completely conserved in all gene clusters, with a few exceptions represented by retroposition events that caused intron loss in some species.

## 4. Discussion

Lysosome biogenesis requires the coordinated expression of genes encoding lysosomal proteins. In mammalian cells, many lysosomal and autophagic genes share one or more 10-base pair DNA motifs in their promoter regions, known as the CLEAR (Coordinated Lysosomal Expression And Regulation) element [46]. Since the CLEAR consensus sequence contains the E-box sequence, it is targeted by members of the MiT/TFE family of basic helix-loop-helix transcription factors [67].

Among them, TFEB and TFE3 directly bind to the CLEAR elements activating the expression of lysosomal genes and facilitating lysosome biogenesis [68,69,70]. A very similar DNA-protein interaction has been observed and demonstrated in *D. melanogaster*, where the Mitf transcription factor plays a major role in regulating V-ATPase activity [8] and lysosomal biogenesis, autophagy, and lipid metabolism [49].

In this work, we have identified in silico the existence of a similar regulatory circuit in a wide range of insects. Orthologous genes encoding V-ATPase subunits and their assembly factors *Tmem199* and *Vma21* in the orders of Diptera, Lepidoptera, Hymenoptera, Coleoptera, Hemiptera, and Phtiraptera contain the dCLEAR element in a conserved position compared to the gene structure observed in *D. melanogaster*. The observed positional constraint of the dCLEAR motif is clearly evident in the Drosophilidae (43 MYA median time divergence [63]), but is also detectable in distant insect species. Our investigation has indeed involved a wide range of informative insect species spanning 550 MY of evolution (*D. melanogaster* vs. Ixodes scapularis, [63]). The Strepsiptera Order apparently constitutes an exception since the dCLEAR element was never found in V-ATPase-encoding genes. A possible explanation can be found in the poor quality of the genome assembly (https://www.ncbi.nlm.nih.gov/assembly/GCA_000281935.1/, accessed on 1 June 2023), which consists of a large number of contigs (94,953) not yet assembled into scaffolds, with a low N50 (4038) and high L50 (10,085) compared to other genomes of similar size or higher. An additional hypothesis is that Strepsiptera have evolved a MITF-independent regulation of the V-ATPase gene expression. Since this is the only representative of the Strepsiptera order that has been sequenced so far, it is not possible to confirm that species of the Strepsiptera Order have escaped from this regulatory system. However, if this exception is confirmed, the absence of the dCLEAR motif will be worth of investigation.

In this work, we have strengthened the concept that the regulation of the coordinated expression of genes encoding molecular complexes is strongly conserved in insects. Evolutionary conservation is often associated with a functional association. In the case of V-ATPase-encoding genes, it has been previously proposed that the presence of the dCLEAR element could be associated with their tissue-specific expression. Indeed, the expression of V-ATPase genes containing dCLEAR in *D. melanogaster* is markedly confined in the salivary gland and the gut of adults and larvae (Figure 5B). A very similar expression pattern is observed in *T. castaneum*, where the dCLEAR-containing V-ATPase-encoding genes are expressed in the gut (Figure 5C). Considering that the median divergence between the flour beetle and the fruit fly is estimated at around 330 Mya [63], this observation strongly argues in favor of the functional conservation of the dCLEAR in conferring a tissue-specific gene expression pattern. Also, this observation further confirms the importance of the V-ATPase complex in epithelia in insects. Unfortunately, due to the lack of extensive gene duplication in *T. castaneum*, it is not possible to evaluate the consequences of the dCLEAR loss. A single V-ATPase-encoding gene duplication event was observed, which involves the *Vha13* (ATP6v1G) gene. The duplicated *Vha13* gene has lost its gut expression while it is more expressed in the brain (Figure 5C).

Another transcriptional network that is possibly controlled and coordinated by a common DNA motif is the mitochondrial OXPHOS system. Nuclear genes encoding proteins of complexes involved in oxidative phosphorylation and the Krebs cycle share a common regulatory DNA, designated as NRG motif [71].

Genome-wide studies have found the NRG in a range of Dipteran and non-Dipteran species [72], not only in OXPHOS-related genes, but also in genes encoding mitochondrial carriers involved in the mitochondrial metabolism [73,74]. Although this has not yet been investigated, it is possible that conserved members of the PAR subfamily of the basic leucin zipper transcription factors [75] can participate in the regulation of the NRG-containing genes [71].

How regulatory circuitry arise is one of the most intriguing evolutionary questions.

One possibility is that the evolution of gene networks is driven by changes in the TFs associated with the expression of specific gene sets. As suggested, duplicated genes encoding TFs followed by mutations in their DNA binding domain can boost the expansion, divergence, and emergence of new regulatory networks through sub-functionalization or neo-functionalization [76].

However, the presence of a conserved DNA motif over multiple and evolutionary divergent species suggests that homologous TFs are involved. The dissemination and the evolutionary selection of DNA motifs at the basis of complex gene networks is also an intricate process that deserves investigation. Given their role in the evolution of genome structure and functions [77,78,79,80], transposable elements can disseminate *cis*-regulatory sequences, as demonstrated by numerous studies (reviewed in [81] and references therein) and also in Drosophila. This evolutionary mechanism has been demonstrated or inferred by several reports (see [82] for a review), an evolutionary aspect also enhanced by the ability of many TEs to perform horizontal transfer [83] even between evolutionarily distant species [84]. However, we have not found insertions of transposable elements or transposition signatures in the sequences analyzed, which suggests that different mechanisms contributed to the duplicated genes’ origin, as observed in previous studies [85]. As an alternative hypothesis, very ancient transposition events that occurred in the ancestor of insects can explain the presence of the dCLEAR motif in the genome of many phylogenetically distant insects, although these events are difficult to be identified in extant species.

In order to understand how the emergence of *cis*-regulatory elements accompanies the changes in the gene network’s structure, it is important to investigate the evolutionary dynamics of these genes and how these changes impact gene expression. Duplication is the main process by which gene diversification occurs and genome size increases over evolutionary time [86,87]. Gene duplication provides the crude genetic material for the evolution of new genes [86]. Once duplicated, the differential evolutionary pressure acting on these sequences is the main force that drives their structural and functional diversification [88].

Retroposition and chromosomal duplications are mechanisms at the basis of gene duplication. These mechanisms have different impacts on diversification since they differ in the persistence (duplication) or removal (retroposition) of upstream regulatory regions and introns.

In the case of the dCLEAR-containing genes, the diversification in structure can be regarded as the main cause of functional diversification, especially when the dCLEAR element is found in an intron or upstream of the TSS.

A potential event involving retroposed genes is gene sub-functionalization. It consists of the partitioning of biological functions between a gene copy and its parental gene, often via a subdivision of the original expression pattern or cellular localization between the two paralogs [89,90,91]. Sub-functionalized paralogs can be observed in almost all *D. melanogaster* V-ATPase gene clusters, as the expression pattern of the dCLEAR-less paralogs is extremely different from their cognate—dCLEAR-bearing—genes (Figure 5A,B). Worth noting, 11 out of 17 paralogs lacking the dCLEAR motif are expressed in the testis, an expression pattern typical of new genes that are in the process of evolving new functions [88]. Also, this observation fits the “out of the testis” hypothesis, which postulates that unique genomic features and powerful selective pressures in the male germline encourage the creation of new genes [91]. In the absence of additional tissue-specific transcriptomic data, it is difficult to infer sub-functionalization in other species, making it only possible to speculate on this issue.

## 5. Conclusions

This study strengthens previous hypotheses regarding the conservation throughout the evolution of a regulative expression circuitry that coordinates the V-ATPase complex and lysosomal biogenesis, which is possibly directed by the same family of evolutionarily conserved transcription factors in different insect species.

This study provides a robust analysis of the evolutionary changes of the V-ATPase subunit-encoding genes that occurred in Arthropoda. While the overall organization of genes has changed over time in terms of exons, introns numbers, and position, the presence of a regulatory motif putatively involved in their mode of expression virtually remains unchanged. The reconstruction of the gene structure in Arthropoda provides evidence of intron loss during evolution. Given the pivotal role of the V-ATPase complex in many biological processes, the presence of a common *cis*-regulatory element could be considered a potential target for genome editing-based approaches to deepen our knowledge of gene expression. Since vacuolar-ATPase is considered a potential candidate target for the control of pests [92], deepening our knowledge of the regulation underlying their expression will possibly aid the development of new control strategies. Finally, further studies based on experimental approaches will be necessary to confirm if the functional role of the dCLEAR motif is conserved in insects, and if our results could also be relevant for a better understanding of human genetic diseases related to lysosomal dysfunctions, since, despite the long evolutionary divergence time, many features of key biological mechanisms are conserved between Drosophila, non-model insects, and humans, and about 70% of the genes associated with human disease have direct counterparts in the Drosophila genome [93,94], as demonstrated by several decades of studies.

## Figures and Tables

**Figure 1 biology-12-01127-f001:**
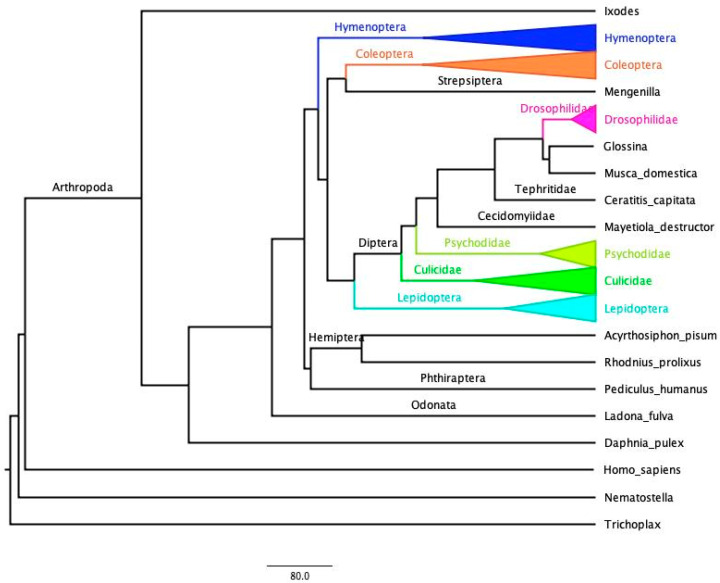
Phylogenetic trees of the species investigated in this study. The tree was obtained from TimeTree [63] and edited with FigTree (http://tree.bio.ed.ac.uk/software/figtree/, accessed on 17 July 2023). Collapsed clades containing multiple species are colored.

**Figure 2 biology-12-01127-f002:**
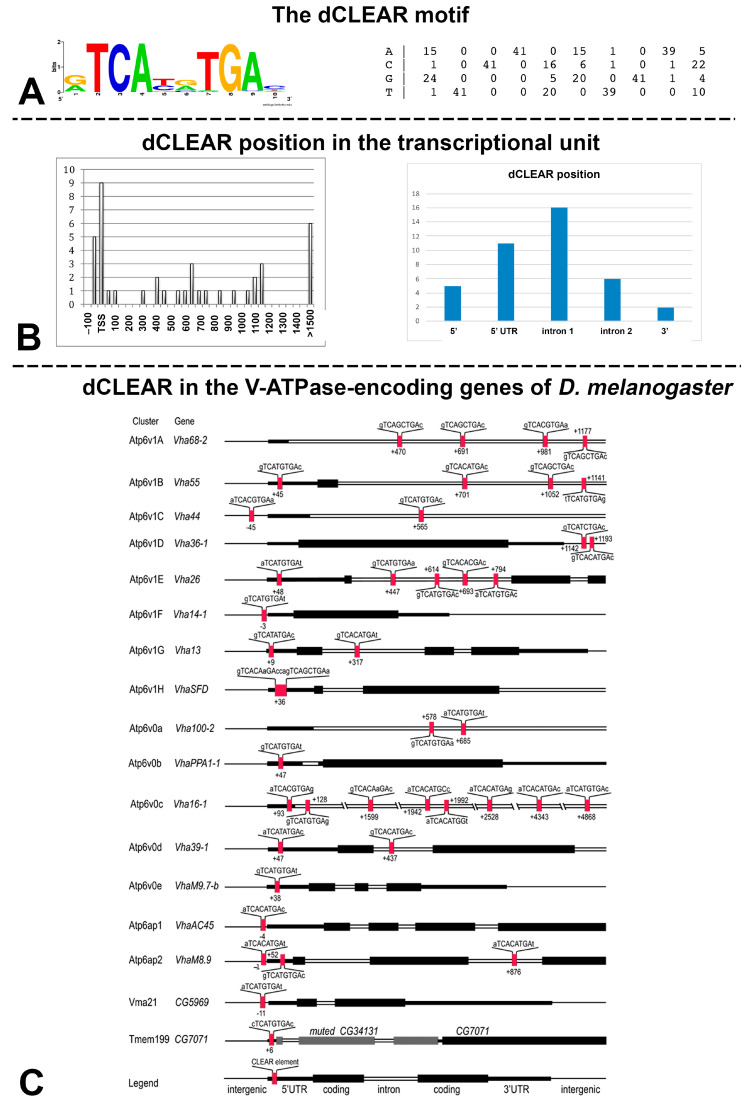
dCLEAR element distribution. (**A**) DNA Logo (left) and Matrix (right) describing the dCLEAR motif in *D. melanogaster*. (**B**) Position analysis of the dCLEAR motif reflecting its position relative to the TSS (left) or to the gene structure (right). The number of occurrences is reported on the *Y*-axis. (**C**) Detailed representation of the dCLEAR motif distribution in each of the 17 genes analyzed in *D. melanogaster*. The legend provides the explanation of black boxes, single lines, and double lines. Exons of the *muted* gene, which is part of a bi-cistronic transcriptional unit and does not encode for a V-ATPase subunit, are depicted in grey.

**Figure 3 biology-12-01127-f003:**
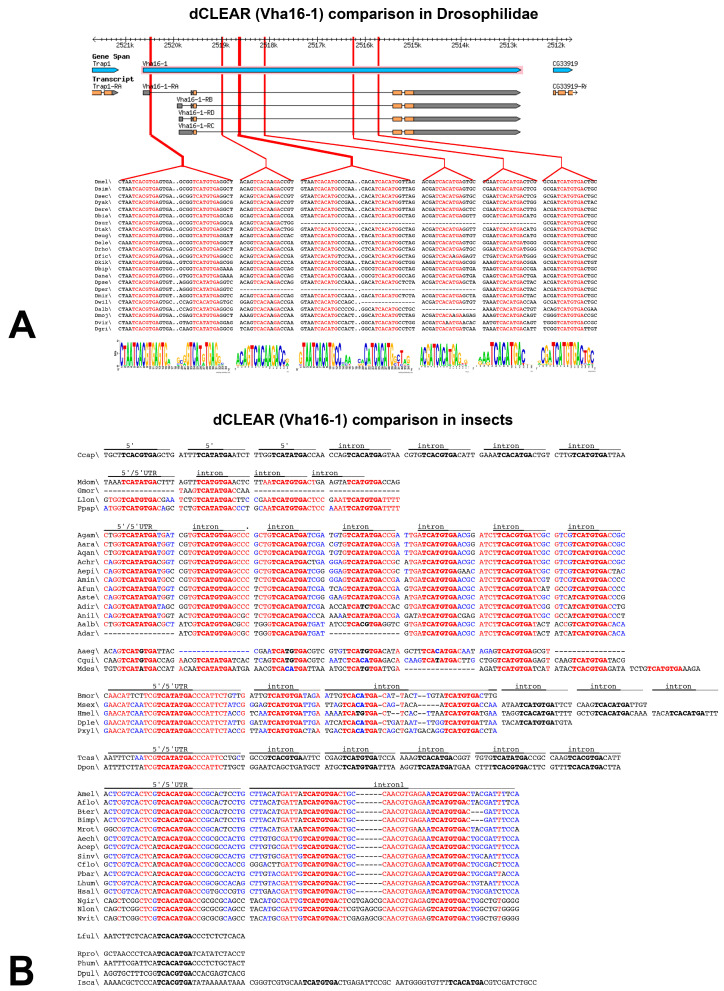
Comparison of the dCLEAR motif. (**A**) Comparison of the dCLEAR motif in orthologous *Vha16-1* genes of Drosophilidae. (**B**) Comparison of the dCLEAR motif in orthologous *Vha16-1* genes of Insect species. Sequences were aligned based on their position within the indicated gene features (5′/5′UTR or intron). The dCLEAR element is in -boldfaced fonts. The high consensus value (90%) is shown in red; the low consensus value (50%) is shown in blue.

**Figure 4 biology-12-01127-f004:**
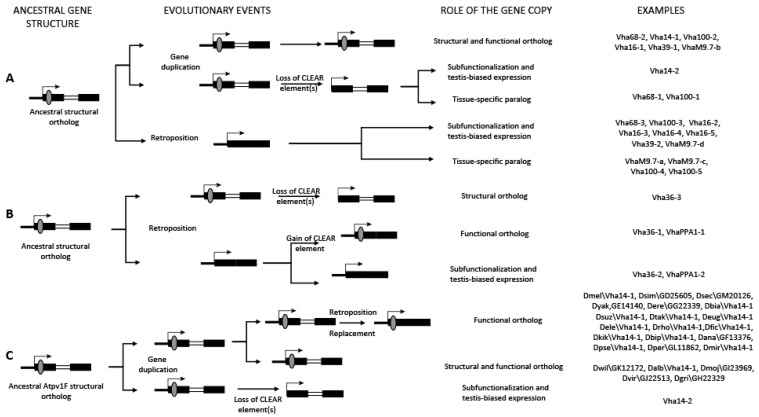
Evolutionary dynamics of dCLEAR gain/loss. Panels (**A**–**C**) depict three different evolutionary scenarios (described in the main text) that describe how genes encoding V-ATPase subunit and regulatory proteins in Drosophilidae originated.

**Figure 5 biology-12-01127-f005:**
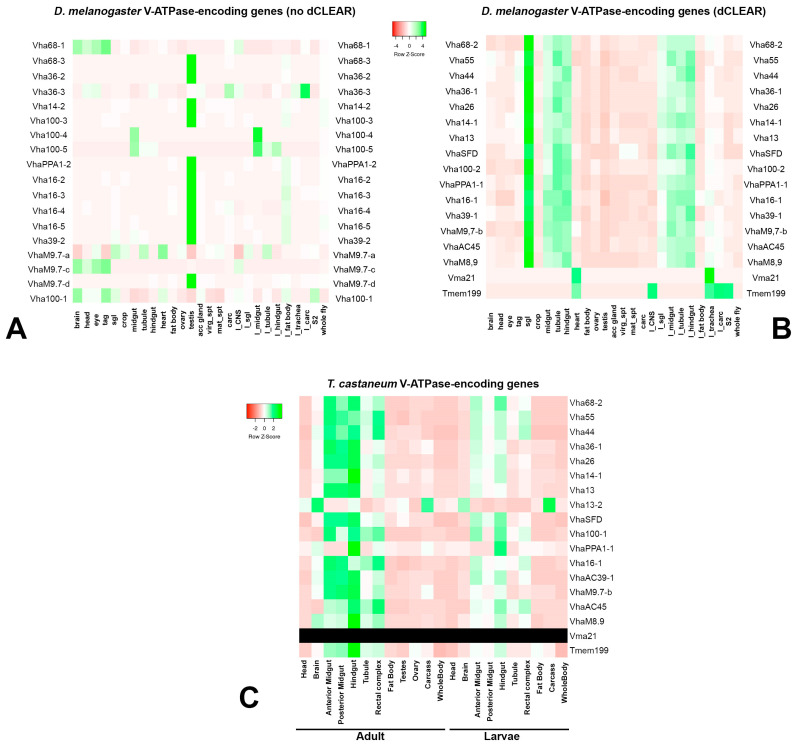
Expression of the V-ATPase genes analyzed in this study. *D. melanogaster* V-ATPase-encoding genes without (**A**) and with (**B**) the dCLEAR motif. Expression data have been retrieved from FlyAtlas [60]. Tissues are displayed in the following order: Brain, Head, Eye, Thoracicoabdominal ganglion, Salivary gland, Crop, Midgut, Tubule, Hindgut, Heart, Fat body, Ovary, Testis, Male accessory gland, Virgin spermatheca, Mated spermatheca, Adult carcass, Larval CNS, Larval Salivary gland, Larval midgut, Larval tubule, Larval hindgut, Larval fat body, Larval trachea, Larval carcass, S2 cells (growing), Whole fly. (**C**) Expression across tissues in adults and larvae of orthologous genes in *T. castaneum*. The *Vma21* ortholog was not found in *T. castaneum* (in black).

**Figure 6 biology-12-01127-f006:**
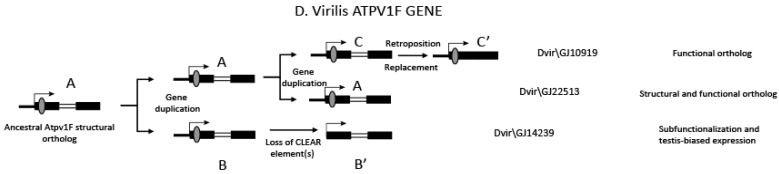
Evolution of the Atpv1F gene structure in *D.virilis*. A. Master copy; B. B,’ C, C’ duplicated copies. The hypothetical mechanisms are indicated.

**Figure 7 biology-12-01127-f007:**
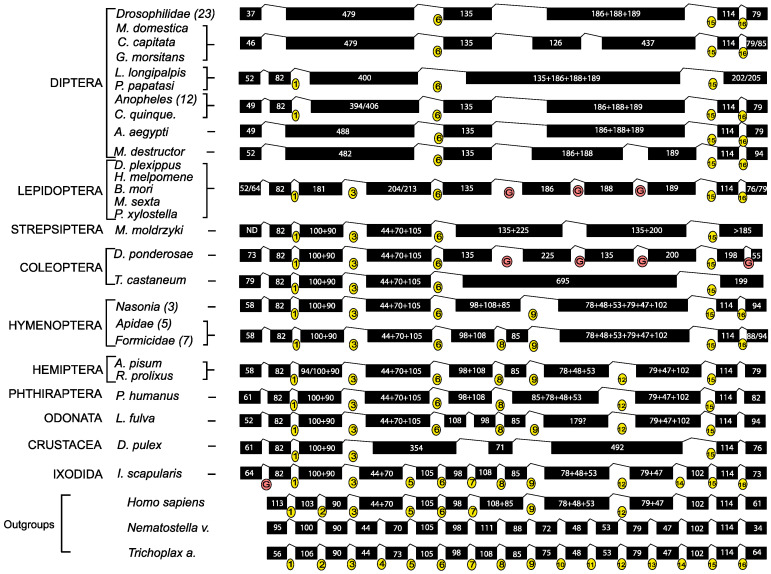
Structural comparison and reconstruction of the evolutionary history of Atp6v1H-encoding gene structure in insect species. Introns are numbered with respect to the *T. adhaerens* gene (yellow circles), considered the ancestral form. Putative intron gains are marked by red circles. The length of exons is indicated by the numbers within black boxes. The length of exons originating from exon fusion (due to intron loss) is represented as the sum of the length of the parental exons.

## Data Availability

All the data are associated with the manuscript submission.

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
