# Peer review of "Structural and Comparative Analyses of Insects Suggest the Presence of an Ultra-Conserved Regulatory Element of the Genes Encoding Vacuolar-Type ATPase Subunits and Assembly Factors"

_biology, 2023, doi:10.3390/biology12081127_

Round 1
Reviewer 1 Report
The current manuscript by Lovero et al. leverages the vast information contained in sequenced insect genomes to analyze orthologous genes encoding subunits of the V-ATPase complex and assembly factors in 70 insect species. By conducting in silico analysis, the authors discovered that orthologous genes encoding V-ATPase subunits and their assembly factors, Tmem199 and Vma21, in various insect orders contain the conserved dCLEAR element in a similar position to that observed in D. melanogaster. The authors find that this positional constraint of the dCLEAR motif is evident not only in closely related Drosophilidae species but also in distantly related insect species, spanning 550 million years of evolution. Together, the comparative analysis reveals a high conservation of the dCLEAR element position, suggesting the existence of an ultra-conserved gene regulation system in insects similar to the CLEAR/TFEB-based regulation system observed in vertebrates.
Overall, I find the subject of the manuscript to be interesting and relevant, and the paper is well-written. The data is presented in a clear way and they fully back the conclusions made. As such, I have no concerns about recommending publication of this manuscript after the authors have addressed my editorial and scientific suggestions, which are detailed below.
Major comments:
Given that this manuscript sets out to analyze insect genomes, and the fact that one of the most important jobs of the H+-V-ATPase is to energize virtually all insect epithelia, there is a striking lack of literature not being mentioned. In particular seminal work by groups led by William Harvey, Helmut Wieczorek, and Julian Dow. As such, work on the epithelial roles of the plasma membrane H+ V-ATPase in controlling transepithelial transport should be included and later on discussed in this paper.
The tissue-specific expression pattern of V-ATPase genes containing/lacking the dCLEAR motif is very intriguing. Do the same tissues show the same composition of dCLEAR-containing genes across insect evolution? An analysis of whether a similar or different patterns can be observed in Tribolium castaneum, an evolutionary distant species to D. melanogaster, should be conducted using the newly published tissue-specific expression atlas BeetleAtlas.org (Naseem et al., 2023. PNAS 120 (13) e2217084120) and could help resolve the evolutionary conservation of the dCLEAR motif in the V-ATPase genes. Such an analysis could help broaden the scope of the findings presented in the manuscript and perhaps help shed new light on the evolution the gene regulatory elements controlling V-ATPase gene expression.
Minor comments:
L25: ”…understanding of regulative..” remove ”of”.
L27: change “analyzes” to “analyzed”
L29: As previously mentioned, given that it is insect genomes that are being analyzed it is important to mention that the V-ATPase is the main pump energizing epithelial transport in insects, and so the cell membrane should also be mentioned here.
L32: Perhaps mention the genes being analyzed?
L50 : “a set” or “sets” of genes.
L87-88: subunits have multiple isoforms?
L106: “of” the V-ATPase coding genes
Figures
Text size in several of the figure should be increased to improve readability.
The general quality of english of the manuscript is good.
Author Response
We thank the reviewer for the positive comments on our manuscript and for the constructive suggestions.
We have expanded the introduction, which now also describes the role of V-ATPase in insect epithelia. We have also cited the important papers suggested by the reviewer.
In order to assess if the evolutionary conservation of the dCLEAR motif is also linked to a conserved function linked to the gene expression pattern of dCLEAR-containing genes, we have queried the BeettleAtlas database, as suggested by reviewer 1. We have summarized the V-ATPase coding gene expression data in an updated version of Figure 5. Thanks to this helpful suggestion we have strengthen the hypothesis that the dCLEAR has a role in conferring a narrow gene expression pattern in the gut. We have commented this result in the discussion.
Many minor issues have been fixed throughout the text.
We have also improved figures quality and captions for a better readability.
Reviewer 2 Report
In this study, Lovero et al., conducted the gene strucctures and tissue-specific expression of V-ATPase subunits and ssembly factors. Overall, the English of this work is pretty well, and it may be of interest for scientifs interested in this topic. My major concerns are:
1) The introduction is a little bit poor. In my opinion, you should provide some examples to tell the readers how important of the genes encoding ATPase subunits and assembly factors. For example, knockdown of these genes casued development defects of Drosophila?
2) Your materials and methods are needed to be improved by providing more detail information.
3) Like your introduction, you should talk about something like the biological functions of genes encoding ATPase subunits and assembly factors. Based on these information, readers can understand why you did this research easily.
Minor comments:
1. You should change your fig.1. In current version, the quality is too low to read.
2. Figure legends of Fig. 2, 3 and 4 should be revised like fig. 5.
3. Your reference formats are also needed to be rechecked carefully.
English of this study is pretty well.
Author Response
Thank you very much for your comments.
As requested, additional information and additional references have been added in the introduction, in order to support of the important role of the V-ATPase complex in insects .
The introduction has been also improved based on the comments of reviewer 1.
We have updated the methods section providing more details.
Figure 1 has been replaced with a new version at higher resolution and we have also revised the figure legends as requested. Other figures have been revised.
Reference have been formatted according to the Journal's guidelines. We have double-checked them carefully.
Round 2
Reviewer 1 Report
The authors have successfully addressed my comments and suggestions and a fully support publication of this manuscript.